

# Diagnostic value and characteristic analysis of serum nucleocapsid antigen in COVID-19 patients

Xihong Zhang[1,2,*], Chungen Qian[3,*], Li Yang[2], Huixia Gao[2], Ping Jiang[4], Muwei Dai[5], Yuling Wang[4], Haiyan Kang[4], Yi Xu[2], Qian Hu[2], Fumin Feng[1], Bangning Cheng[6] and Erhei Dai[2]

[1] School of Public Health, North China University of Science and Technology, Tangshan, Hebei, China
[2] Department of Laboratory Medicine, The Fifth Hospital of Shijiazhuang, North China University of Science and Technology, Shijiazhuang, Hebei, China
[3] The Key Laboratory for Biomedical Photonics of MOE at Wuhan National Laboratory for Optoelectronics-Hubei Bioinformatics & Molecular Imaging Key Laboratory, Systems Biology Theme, Department of Biomedical Engineering, College of Life Science and Technology, Huazhong University of Science and Technology, Wuhan, Hubei, China
[4] Department of Tuberculosis, The Fifth Hospital of Shijiazhuang, North China University of Science and Technology, Shijiazhuang, Hebei, China
[5] Orthopaedic Department, The Fourth Hospital of Hebei Medical University and Hebei Cancer Hospital, Shijiazhuang, Hebei, China
[6] Shenzhen YHLO Biotech Co., Ltd, Shenzhen, Guangdong, China
* These authors contributed equally to this work.

Corresponding authors
Bangning Cheng,
chengbangning@szyhlo.com
Erhei Dai, daieh2008@126.com

## ABSTRACT

**Background:** To date, several types of laboratory tests for coronavirus disease 2019 (COVID-19) diagnosis have been developed. However, the clinical importance of serum severe acute respiratory syndrome coronavirus 2 (SARS-CoV-2) nucleocapsid antigen (N-Ag) remains to be fully elucidated. In this study, we sought to investigate the value of serum SARS-CoV-2 N-Ag for COVID-19 diagnosis and to analyze N-Ag characteristics in COVID-19 individuals.

**Methods:** Serum samples collected from 215 COVID-19 patients and 65 non-COVID-19 individuals were used to quantitatively detect N-Ag *via* chemiluminescent immunoassay according to the manufacturer's instructions.

**Results:** The sensitivity and specificity of the N-Ag assay were 64.75% (95% confidence interval (95% CI) [55.94–72.66%]) and 100% (95% CI [93.05–100.00%]), respectively, according to the cut-off value recommended by the manufacturer. The receiver operating characteristic (ROC) curve showed a sensitivity of 100.00% (95% CI [94.42–100.00%]) and a specificity of 71.31% (95% CI [62.73–78.59%]). The positive rates and levels of serum SARS-CoV-2 N-Ag were not related to sex, comorbidity status or disease severity of COVID-19 (all $P < 0.001$). Compared with RT-PCR, there was a lower positive rate of serum N-Ag for acute COVID-19 patients ($P < 0.001$). The positive rate and levels of serum SARS-CoV-2 N-Ag in acute patients were significantly higher than those in convalescent patients (all $P < 0.001$). In addition, the positive rate of serum SARS-CoV-2 N-Ag in acute COVID-19 patients was higher than that of serum antibodies (IgM, IgG, IgA and neutralizing antibodies (Nab)) against SARS-CoV-2 (all $P < 0.001$). However, the positive rate of

serum SARS-CoV-2 N-Ag in convalescent COVID-19 patients was significantly lower than that of antibodies (all $P < 0.001$).

**Conclusion:** Serum N-Ag can be used as a biomarker for early COVID-19 diagnosis based on appropriate cut-off values. In addition, our study also demonstrated the relationship between serum N-Ag and clinical characteristics.

## INTRODUCTION

Coronavirus disease 2019 (COVID-19) is an acute respiratory infectious disease caused by severe acute respiratory syndrome coronavirus 2 (SARS-CoV-2) infection. Some COVID-19 patients may present with asymptomatic or mild symptoms (*e.g.*, cough, fever or malaise), but severe cases can manifest as acute respiratory distress syndrome, multiorgan failure or even death (*Huang et al., 2020*; *Wang et al., 2020a*). Rapid and effective identification of SARS-CoV-2 infection represents the global response to COVID-19, which allows for infected individuals to be treated promptly and helps to control the transmission of SARS-CoV-2. COVID-19 diagnosis primarily depends on viral RNA detection by using real-time reverse transcription polymerase chain reaction (RT-PCR), specific antibody testing and specific antigen testing (*Peeling et al., 2022*; *Filchakova et al., 2022*).

RT-PCR represents the gold standard technique for COVID-19 diagnosis with high sensitivity and specificity (*Bohn et al., 2020*; *LeBlanc et al., 2020*; *Corman et al., 2020*; *Chu et al., 2020*). However, the detection sensitivities of RT-PCR methods can be dependent on the detection gene region of SARS-CoV-2 (*Corman et al., 2020*; *Nalla et al., 2020*) and different types of detection samples (*Wang et al., 2020b*); moreover, RT-PCR requires the prolonged use of expensive equipment in special operating locations to obtain results. In the early stage of the COVID-19 pandemic, specific antibody testing was a useful diagnostic tool for those who failed to detect SARS-CoV-2 *via* RT-PCR-based testing (*Deeks et al., 2020*). However, with the spread of COVID-19 vaccinations, antibodies can no longer be used as a specific indicator for COVID-19 diagnosis. Therefore, there is still a need for more effective and sensitive diagnostic methods for COVID-19 diagnosis.

Some rapid antigen detection methods based on respiratory samples have been applied and could have comparable performance to RT-PCR assays, with satisfactory sensitivity ranging from 75.6% to 100% and specificity ranging from 75.8% to 100% (*Chaimayo et al., 2020*; *Mayanskiy et al., 2021*; *Diao et al., 2021*; *Pollock et al., 2021*; *Cirit et al., 2023*). However, the quality and quantity of respiratory specimens are difficult to standardize compared with blood samples. Accordingly, blood antigen biomarkers may have inherent advantages over upper respiratory antigen testing or RT-PCR. Additionally, there is growing evidence that the detection of viral proteins in the blood may represent a new approach to improve the screening or diagnosis of SARS-CoV-2 infection (*Mathur et al., 2022*; *Verkerke et al., 2022*; *Wang et al., 2021*; *Zhang et al., 2021*; *Li et al., 2020*; *Hingrat*

*et al., 2020*). Previous studies have also reported that decreased nucleocapsid antigen (N-Ag) levels in the blood were significantly associated with increased anti-SARS-CoV-2 antibodies (*Zhang et al., 2021*; *Shan et al., 2021*; *Costa et al., 2022*; *Oueslati et al., 2022*). In addition, several studies have shown that N-Ag levels in the blood are closely correlated with disease severity, as reflected by intensive care unit (ICU) admission and grade of disease severity in COVID-19 patients (*Wang et al., 2021*; *Zhang et al., 2021*; *Oueslati et al., 2022*; *Favresse et al., 2022*). Nucleocapsid protein (NP) is the most abundant structural protein of SARS-CoV-2, which packages RNA to form a helical nucleocapsid (*Masters, 2019*). Moreover, NP plays an important role in viral mRNA transcription and replication (*Cong et al., 2020*). In particular, NP can induce cellular and humoral immune responses after SARS-CoV-2 infection (*Ni et al., 2020*). Based on the abovementioned findings, the examination of N-Ag in the blood remains of great importance.

In this study, we evaluated the diagnostic value of N-Ag through a rapid diagnostic test and analyzed N-Ag characteristics in COVID-19 patients. The diagnostic test, which is known as the iFlash-2019-nCoV Antigen Assay (Shenzhen YHLO Biotech Co., Ltd, Shenzhen, China), is a paramagnetic particle chemiluminescence immunoassay for the qualitative detection of SARS-CoV-2 N-Ag in nasopharyngeal and nasal swab samples. In a previous study, the iFlash-2019-nCoV Antigen Assay was also used to detect SARS-CoV-2 N-Ag in serum, and good sensitivity (76.27%) and specificity (98.78%) were obtained (*Deng et al., 2021*). Therefore, we used this assay to detect serum SARS-CoV-2 N-Ag in the current study.

## MATERIALS AND METHODS

### Study design, patients, and ethics

This study was an observational study conducted in the Fifth Hospital of Shijiazhuang (which is a tertiary referral university hospital located in Hebei province in China, which mainly treats patients with various infectious diseases such as COVID-19 and infectious liver diseases). We enrolled 215 COVID-19 patients and 65 non-COVID-19 individuals from January to February 2021, and all of the participants were older than 14-years-old. The diagnosis of COVID-19 and the evaluation of the grade of disease severity were based on the results of RT–PCR and pathological changes observed *via* computed tomography images, according to China's Diagnosis and Treatment Protocol for COVID-19 (8th trial edition). All of the individuals were classified as asymptomatic, mild, moderate, severe and critical cases. Remarkably, all of the participants in this study had not been vaccinated against SARS-CoV-2. This study was approved by the Ethics Committee of the Fifth Hospital of Shijiazhuang (protocol number 2020008). The need for informed consent was waived for all of the participants.

### Clinical specimens

A total of 281 serum samples collected from 215 individuals were classified into 122 acute-phase sera and 159 convalescent-phase sera. The acute-phase sera were defined as follows: sera from confirmed cases within 7 days after illness onset or asymptomatic cases within 7 days after initial positive SARS-CoV-2 RNA. The convalescent-phase sera were

defined as follows: sera from individuals who met the criteria and who were discharged from the hospital, more than 14 days after illness onset or the initial positive SARS-CoV-2 RNA. The discharge criteria complied with China's Diagnosis and Treatment Protocol for COVID-19 (8th trial Edition). Respiratory samples for RT-PCR were obtained from participants' nasopharyngeal swabs, with a sampling interval of less than 48 h between nasopharyngeal swabs and serum.

## Sample detection

An RNA extraction kit (DA AN Gene, Guangzhou, China) was used to extract SARS-CoV-2 RNA from participants' nasopharyngeal swabs, and SARS-CoV-2 RNA was detected by using ABI Prism 7500 *via* a SARS-CoV-2 RNA detection kit (DaAn Gene, Guangzhou, China). The samples with RT-PCR cycle threshold values above 40 were considered to be negative according to the manufacturer's recommendations. SARS-CoV-2 antigen (catalog number: C86096) and antibody (catalog numbers of IgM, IgG, IgA and neutralization antibody (NAb) were C86096, C86095M, C86095G and C86109, respectively) assay reagent kits (YHLO Biotech Co, Ltd, Shenzhen, China) were used to perform a paramagnetic particle chemiluminescent immunoassay for the quantitative detection of serum N-Ag and antibodies (including IgM, IgG, IgA and NAb) by using an iFlash3000-C Chemiluminescence Immunoassay Analyzer (Shenzhen YHLO Biotech Co, Ltd, Shenzhen, China). The positivity threshold was set to 1 COI and 10 AU/mL, respectively, as recommended by the manufacturer. All of the testing procedures were performed in accordance with the manufacturer's instructions.

## Statistical analysis

All of the statistical analyses and figure drawings were performed by using GraphPad Prism software 9.0 (GraphPad, San Diego, CA, USA). Continuous variables were presented as the medians and interquartile ranges (IQRs). Independent continuous variables were compared by using the Mann-Whitney U test. Categorical variables were presented as frequencies and percentages and were compared by using the chi-square test or Fisher's exact test, as appropriate. A $P$ value of less than 0.05 was considered to be statistically significant.

In addition, to assess the performance characteristics of serum SARS-CoV-2 N-Ag against affirmed infection (gold standard), a receiver operating characteristic (ROC) curve was constructed to illustrate sensitivity, specificity and Youden index (sensitivity + specificity – 1), reported as 95% confidence intervals (95% CIs).

# RESULTS

## Baseline characteristics

The demographics and clinical characteristics of all of the COVID-19 patients and non-COVID-19 individuals were summarized in Table 1. The COVID-19 patients included 75 (34.88%) males and 140 (65.12%) females, with a median age of 49.00 years (IQR: 33.00–60.00 years). Among 215 patients, 59 (27.44%) had comorbidities, and hypertension (20.47%) was the most common comorbidity. There were 54 asymptomatic patients

**Table 1 The participants' demographics and clinical characteristics.**

| Characteristic | COVID-19 patients (N = 215) | Non-COVID-19 individuals (N = 65) |
|---|---|---|
| Gender | | |
| Male | 75 (34.88%) | 31 (47.69%) |
| Female | 140 (65.12%) | 34 (52.31%) |
| Age, years | 49.00 (33.00, 60.00) | 34.00 (31.00, 55.00) |
| Comorbidity | 59 (27.44%) | – |
| Hypertension | 44 (20.47%) | – |
| Diabetes | 20 (9.30%) | – |
| Cardiovascular disease | 8 (3.72%) | – |
| Cerebrovascular disease | 6 (2.79%) | – |
| Chronic lung disease | 2 (0.93%) | – |
| Chronic nephrosis | 3 (1.40%) | – |
| Chronic liver disease | 4 (1.86%) | – |
| Cancer | 6 (2.79%) | – |
| Grades of disease severity | | |
| Asymptomatic | 54 (25.12%) | – |
| Mild | 8 (3.72%) | – |
| Moderate | 145 (67.44%) | – |
| Severe | 8 (3.72%) | – |

Notes:
The information on non-COVID-19 individuals' comorbidity was not collected at the time.
COVID-19, coronavirus disease 2019.

(25.12%) and 161 (74.88%) symptomatic patients (including eight mild patients (3.72%), 145 moderate patients (67.44%) and eight severe patients (3.72%)). Non-COVID-19 individuals were aged 34.00 years (IQR: 31.00–55.00 years), and 31 (47.69%) were male.

### Diagnostic performance of serum N-Ag for acute COVID-19 samples

Using the confirmed infection as the reference, the diagnostic performance of serum N-Ag for acute COVID-19 samples ($N = 122$) was evaluated. The receiver operating characteristic (ROC) curve demonstrated that the area under the curve (AUC) was 88.68% (95% CI [84.08–93.27%]; $P < 0.001$), and the sensitivity and specificity were 100.00% (95% CI [94.42–100.00%]) and 71.31% (95% CI [62.73–78.59%]), respectively (Fig. 1). The best cut-off value was 0.71 COI. However, according to the cut-off value (1 COI) recommended by the manufacturer, all serum N-Ag levels of non-COVID-19 samples ($N = 65$) were below the detection limit, thus resulting in a specificity of 100% (93.05–100.00%). Among 122 serum samples obtained from acute COVID-19 patients, 79 cases tested positive for SARS-CoV-2 N-Ag, which demonstrated a sensitivity of 64.75% (95% CI [55.94–72.66%]).

### Serum SARS-CoV-2 N-Ag characteristics in acute COVID-19 patients

We analyzed the positive rates and median levels of serum N-Ag in COVID-19 patients with different clinical characteristics. The results showed that there was no statistical significance for the comparison of serum N-Ag among different sexes ($P = 0.24$ and

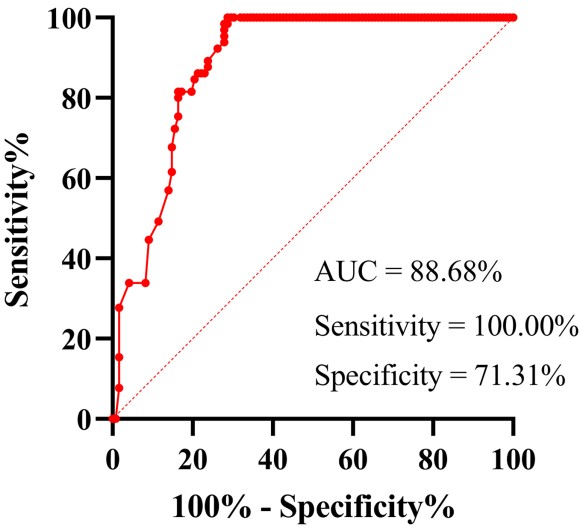

**Figure 1 ROC curve of serum SARS-COV-2 N-Ag.**

$P = 0.09$, respectively), comorbidity statuses ($P = 0.54$ and $P = 0.19$, respectively) and different grades of disease severity ($P = 0.09$ and $P = 0.85$, respectively) (Fig. 2).

## Comparison of serum N-Ag and nasopharyngeal swab SARS-CoV-2 RNA in acute COVID-19 patients

Compared with nasopharyngeal swab SARS-CoV-2 RNA, there was a lower positive rate of serum SARS-CoV-2 N-Ag in acute COVID-19 patients ($P < 0.001$) (Fig. 3).

## Comparison of serum N-Ag between acute and convalescent COVID-19 patients

The positive rate and levels of serum SARS-CoV-2 N-Ag in acute patients were significantly higher than those in convalescent patients (all $P < 0.001$) (Fig. 4).

## Comparison of positive rates of serum N-Ag and antibodies

In addition, serum IgM, IgG, IgA and NAb were detected in serum samples from acute and convalescent COVID-19 patients. The results showed a significantly higher positive rate of serum SARS-CoV-2 N-Ag in acute COVID-19 patients compared with serum antibodies (IgM, IgG, IgA and NAb) (all $P < 0.001$). In contrast, the positive rate of serum N-Ag was significantly lower in convalescent COVID-19 patients than that of serum antibodies ($P < 0.001$) (Fig. 5).

## DISCUSSION

In the context of the COVID-19 pandemic, appropriate diagnostic tests are important to limit the spread of SARS-CoV-2 and to properly manage COVID-19 patients. Although RT‒PCR is considered to be the gold standard for confirming SARS-CoV-2 infection, the performance of an RT‒PCR assay is time-consuming and requires skilled laboratory personnel and special detection equipment. Thus, some rapid diagnostic tests are necessary to rapidly identify SARS-CoV-2 infections, although some of the tests are less sensitive

Peer

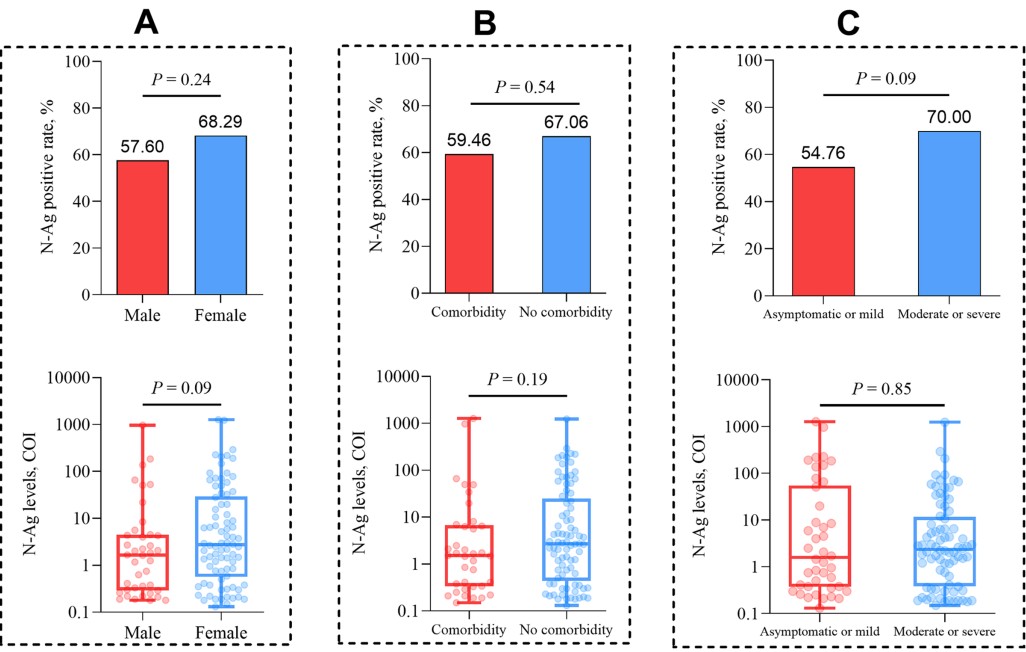

**Figure 2 Comparison of the quantitative and qualitative N-Ag characteristics in acute COVID-19 patients.** (A) Comparison of serum N-Ag in patients of different sexes. (B) Comparison of serum N-Ag in patients with and without comorbidity. (C) Comparison of serum N-Ag in patients of different grades of disease severity. Statistical significance was calculated by the Mann-Whitney U test or the chi-square test. Abbreviations: N-Ag, nucleocapsid antigen.

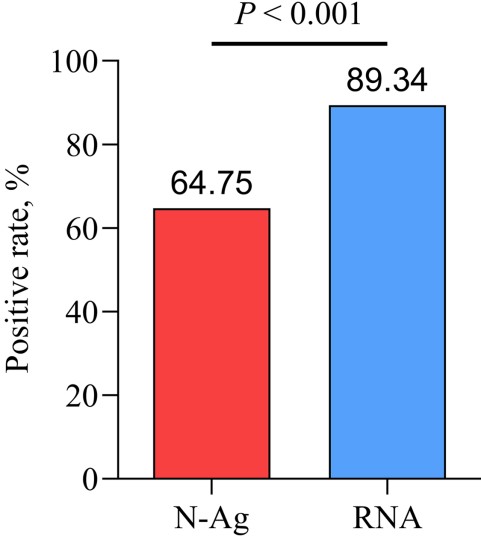

**Figure 3 Comparison of serum N-Ag and nasopharyngeal swab SARS-CoV-2 RNA.** Statistical significance was calculated by the chi-square test. Abbreviations: N-Ag, nucleocapsid antigen.

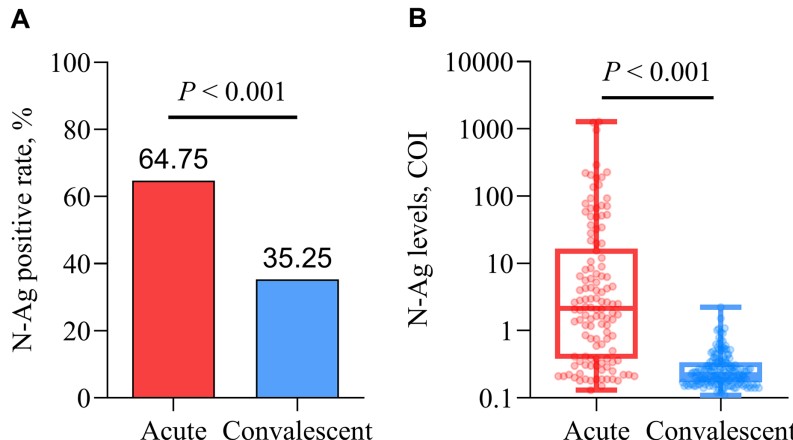

**Figure 4 Comparison of qualitative (A) and quantitative (B) serum N-Ag in acute and convalescent COVID-19 patients.** Statistical significance was calculated by the Mann-Whitney U test or the chi-square test. Abbreviations: N-Ag, nucleocapsid antigen.

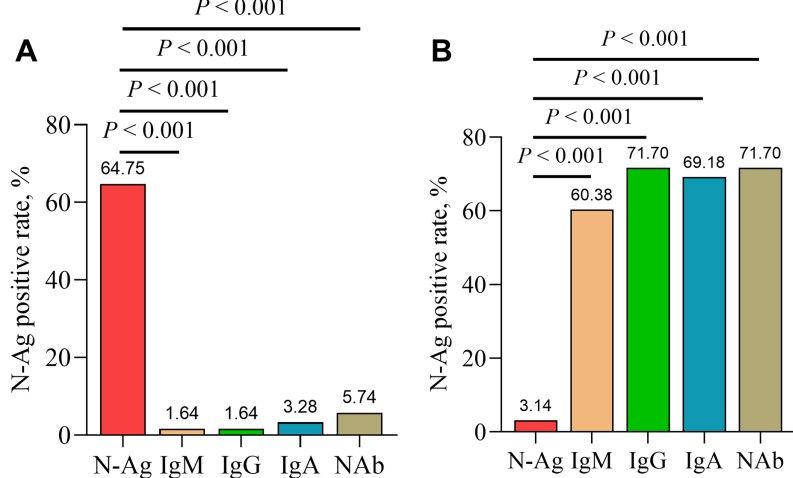

**Figure 5 Comparison of positive rates of serum N-Ag and antibodies in acute (A) and convalescent (B) COVID-19 patients.** Statistical significance was calculated by the chi-square test. Abbreviations: N-Ag, nucleocapsid antigen.

than RT-PCR (*Hirotsu et al., 2020*; *Ciotti et al., 2021*; *Scohy et al., 2020*). Previous studies have demonstrated that SARS-CoV-2 N-Ag can be detected in the blood (including serum and plasma samples) of COVID-19 patients (*Ogata et al., 2020*; *Ahava et al., 2022*; *Su et al., 2021*; *Thudium et al., 2021*; *Olea et al., 2021*). SARS-CoV-2 N-Ag detection in the blood not only provides a new diagnostic biomarker for COVID-19 but also contributes to identifying the relationship with disease severity (*Zhang et al., 2021*; *Li et al., 2020*; *Hingrat et al., 2020*; *Oueslati et al., 2022*). Therefore, it is still of clinical significance to explore the diagnostic value and characteristics of serum N-Ag in COVID-19 patients.

Our study reconfirmed that SARS-CoV-2 N-Ag can be detected in the blood of COVID-19 patients. According to the cut-off value (1 COI) recommended by the manufacturer, the specificity of the serum N-Ag assay was 100%, but the sensitivity only
reached 64.75%. *Yasmin et al. (2022)* tested nasopharyngeal swab N-Ag in COVID-19 patients by using the iFlash-2019-nCoV Antigen Assay and the same cut-off values (1 COI), and the specificity (100%) and sensitivity (64.52%) were similar to our results, which suggests that this antigen assay may have the same detection performance for nasopharyngeal swabs or blood. It has to be mentioned that according to the manufacturer's instruction, the iFlash-2019-nCoV Antigen Assay was originally developed to confirm SARS-CoV-2 infection by detecting nasopharyngeal swab N-Ag. However, based on the above statements, the antigen assay has similar sensitivity (64.75% *vs* 64.52%) and specificity (100% *vs* 100%) for detecting SARS-CoV-2 N-Ag in nasopharyngeal swabs and sera, which supports the idea that the antigen assay can also be used to detect serum SARS-CoV-2 N-Ag.

Here, we note that only two studies have used the iFlash-2019-nCoV Antigen Assay to detect serum SARS-CoV-2 N-Ag. A study from China (Wuhan City) have shown that a higher sensitivity (76.27%) within the first week after symptom onset and a lower sensitivity (62.50%) within the second week after symptom onset, according to the cut-off value of 1.46 COI (*Deng et al., 2021*). Overall, the results of this study were not very different from our results, although the two diagnostic tests used different cut-off values. Notably, the blood samples detected in this study were collected from patients in the early stages of the COVID-19 outbreak in China, so it is certain that they detected the N protein of SARS-CoV-2 wild-type (WT) (Wuhan-Hu-1) strain. However, the N protein of the Alpha variant was detected in our study, which suggests that the iFlash-2019-nCoV Antigen Assay may have the same diagnostic value for different SARS-CoV-2 strains. In another study from Japan (*Yokoyama et al., 2021*), a sensitivity of 91.0% and a specificity of 81.3% based on a cut-off value of 0.255 was demonstrated. This sensitivity is significantly higher than that of our study (91.0% *vs* 64.52%), and they used a relatively low cut-off value that might explain differences in detection. In addition, several studies have reported the diagnostic performance of serum N-Ag with sensitivity (62–92%) and specificity (96.8–100%) (*Perna et al., 2021*; *Hingrat et al., 2020*; *Li et al., 2020*), and the sensitivity (93.3–100%) and specificity (94.8–100%) of nasopharyngeal swab N-Ag were more satisfactory (*Orsi et al., 2021*; *Pollock et al., 2021*; *Pérez-García et al., 2021*; *Gili et al., 2021*). The excellent specificity that was previously reported is generally consistent with our findings, thus suggesting that the serum SARS-CoV-2 N-Ag test may be ideal for confirmatory testing in special situations (for example, blood bank sample testing).

When using the ROC curve to establish the cut-off value (0.71 COI), the sensitivity of the serum N-Ag assay was 100.00%; however, the specificity decreased to 71.31%. Consequently, we suggest that the appropriate cut-off value should be selected according to the user's purpose. The cut-off value obtained from the ROC curve should be recommended if the aim is to perform a viable screening test to identify more SARS-CoV-2-infected individuals. However, if satisfactory specificity is needed, the cut-off value recommended by the manufacturer is recommended. Notably, the iFlash-2019-nCoV Antigen Assay does not differentiate between SARS-CoV and SARS-CoV-2 according to the manufacturer's instructions. Therefore, the results of serum N-Ag detection should be

treated with caution; additionally, if necessary, RT–PCR or specific antibody testing is required to more accurately confirm the diagnosis of SARS-CoV-2 infection.

When compared with the results of nasopharyngeal swab RT–PCR testing, there was a significantly lower positive rate of serum N-Ag in acute COVID-19 patients. This result was understandable, as RT–PCR testing remains the standard diagnostic method for SARS-CoV-2 infections. However, the detection of serum SARS-CoV-2 N-Ag in 13 RT–PCR-negative samples showed that 5 (38.5%) of them were positive in our study. These patients with positive SARS-CoV-2 N-Ag represent infected patients with SARS-CoV-2 diagnoses that may have been missed. The study demonstrated that the viral load in patients' throat swabs was highest at the time of symptom onset and inferred that SARS-CoV-2 infectivity peaked at or before the onset of symptoms (*He et al., 2020*), which creates difficulties for the early diagnosis of SARS-CoV-2 infection. Moreover, false-negative RT–PCR results are a major challenge for confirming SARS-CoV-2 infections (*Kucirka et al., 2020*; *Woloshin, Patel & Kesselheim, 2020*). Therefore, our findings demonstrated the potential role of screening for SARS-CoV-2 N-Ag in people who already have blood drawn but for who the RT–PCR result is negative. Besides, the detection of blood samples has several inherent advantages over the detection of nasopharyngeal swabs. Above all, the quality and quantity of blood specimens collected are more easily standardized than nasopharyngeal swabs. Secondly, confirming SARS-CoV-2 infection by collecting individuals' venous blood may greatly reduce the risk of SARS-CoV-2 infection for healthcare workers. Due to the process of collecting nasopharyngeal swabs by healthcare workers may irritate the subject's throat, leading to coughing and nausea reflexes, which may increase the risk of SARS-CoV-2 infection to the sampler, especially if protective equipment is not used properly (*Paris et al., 2022*). Finally, in some special cases, blood antigen testing play an important role. For example, in the epidemiological investigation, it is found that some of the blood bank samples are from SARS-CoV-2-infected individuals, then it is possible to find the target samples through serum antigen detection.

Previous studies have shown that SARS-CoV-2 viral load in the blood was positively correlated with the severity of COVID-19 (*Hogan et al., 2021*; *Fajnzylber et al., 2020*; *Eberhardt et al., 2020*; *Kawasuji et al., 2022*); however, whether such a relationship exists between blood antigens and disease severity in COVID-19 patients is a question worth exploring. In this study, we compared serum SARS-CoV-2 N-Ag levels and positivity rates between asymptomatic or mild patients and moderate or severe patients. We found that there was no significant difference in the serum N-Ag level and positive rate between the two groups. However, previous studies that differ from our findings suggest that higher serum SARS-CoV-2 N-Ag levels or positive rates are highly correlated with more severe grades of COVID-19 (*Verkerke et al., 2022*; *Yokoyama et al., 2021*; *Deng et al., 2021*), which may be the result of different diagnostic assays and individual differences. *Tsverava et al. (2022)* demonstrated that antibody responses against SARS-CoV-2 NP were significantly higher in women in the convalescent phase than in men, which suggests that COVID-19 patient blood antigens may be associated with sex. However, our study demonstrated no significant differences in positive rates and levels of serum SARS-CoV-2 N-Ag between

females and males, and the reason for this result may be that we only compared serum N-Ag in acute patients rather than in convalescent patients. Furthermore, SARS-CoV-2 RNAemia is more likely to be detected in patients with comorbidities (*Berastegui-Cabrera et al., 2021*); however, it is not known whether comorbidity statuses influence blood antigens. This query was answered in our study, wherein there were no significant differences in SARS-CoV-2 N-Ag positivity rates and levels between patients with or without comorbidities.

In this study, the positive rates and levels of serum SARS-CoV-2 N-Ag in acute COVID-19 samples were significantly higher than those in convalescent COVID-19 samples, which indicated that serum SARS-CoV-2 N-Ag gradually decreased as the disease recovered. Other studies have determined conclusions that were consistent with our findings (*Verkerke et al., 2022*; *Yokoyama et al., 2021*). Furthermore, serum SARS-CoV-2 N-Ag does appear to correlate with antibody seroconversion (*Zhang et al., 2021*; *Shan et al., 2021*; *Costa et al., 2022*; *Oueslati et al., 2022*). We observed that the positive rates of serum SARS-CoV-2 N-Ag in acute COVID-19 patients were higher than those of serum antibodies (including IgM, IgG, IgA and NAb), and there was an opposing result in convalescent patients.

Our study also had several limitations. Above all, there was poor representation due to the limited number of participants who were enrolled. Second, serial sampling was not performed due to limited conditions, thus leading to the result that the longitudinal characteristics of SARS-CoV-2 N-Ag could not be analyzed. Finally, our samples were obtained from COVID-19 patients infected with the alpha variant, and none of the individuals had been vaccinated against COVID-19, which was not representative of the current status of SARS-CoV-2 infection. Consequently, a prospective cohort study with a large sample is necessary to validate our findings in the context of the Omicron variant epidemic.

## CONCLUSIONS

In summary, our study validates that serum SARS-CoV-2 N-Ag is a sensitive and specific biomarker in acute SARS-CoV-2 infection based on appropriate cut-off values. Additionally, the positive rate and level of serum SARS-CoV-2 N-Ag were not related to sex, comorbidity status or disease severity of COVID-19. The positive rate and levels of SARS-CoV-2 N-Ag in acute COVID-19 patients are higher than those in convalescent COVID-19 patients. Furthermore, serum SARS-CoV-2 N-Ag is reduced with antibody seroconversion.

## ACKNOWLEDGEMENTS

We thank all people who participated in the study.

### Funding

This research was funded by the Key Research and Development Program of Hebei province (No. 22377744D) and the Research and Development Program of Shijiazhuang Science and Technology (No. 211200443). The funders had no role in study design, data collection and analysis, decision to publish, or preparation of the manuscript.

### Grant Disclosures

The following grant information was disclosed by the authors:
Key Research and Development Program of Hebei province: 22377744D.
Research and Development Program of Shijiazhuang Science and Technology: 211200443.

### Competing Interests

Bangning Cheng is employed by Shenzhen YHLO Biotech Co., Ltd, and provided detection reagents for our study.

### Author Contributions

- Xihong Zhang conceived and designed the experiments, analyzed the data, prepared figures and/or tables, authored or reviewed drafts of the article, and approved the final draft.
- Chungen Qian conceived and designed the experiments, analyzed the data, authored or reviewed drafts of the article, and approved the final draft.
- Li Yang performed the experiments, analyzed the data, authored or reviewed drafts of the article, and approved the final draft.
- Huixia Gao performed the experiments, analyzed the data, prepared figures and/or tables, and approved the final draft.
- Ping Jiang performed the experiments, analyzed the data, prepared figures and/or tables, and approved the final draft.
- Muwei Dai performed the experiments, prepared figures and/or tables, collection of basic information about participants, and approved the final draft.
- Yuling Wang performed the experiments, prepared figures and/or tables, authored or reviewed drafts of the article, collection of basic information about participants, and approved the final draft.
- Haiyan Kang performed the experiments, prepared figures and/or tables, and approved the final draft.
- Yi Xu performed the experiments, prepared figures and/or tables, and approved the final draft.
- Qian Hu performed the experiments, prepared figures and/or tables, and approved the final draft.
- Fumin Feng conceived and designed the experiments, authored or reviewed drafts of the article, and approved the final draft.

- Bangning Cheng conceived and designed the experiments, authored or reviewed drafts of the article, and approved the final draft.
- Erhei Dai conceived and designed the experiments, authored or reviewed drafts of the article, and approved the final draft.

### Human Ethics

The following information was supplied relating to ethical approvals (*i.e.*, approving body and any reference numbers):

The study was approved by the Ethics Committee of the Fifth Hospital of Shijiazhuang (2020008).

### Data Availability

The raw measurements are available in the Supplemental Files.

### Supplemental Information

Supplemental information for this article can be found online at http://dx.doi.org/10.7717/peerj.15515#supplemental-information.

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
