# Peer review of "Diagnostic value and characteristic analysis of serum nucleocapsid antigen in COVID-19 patients"

_PeerJ, doi:10.7717/peerj.15515_

## Round 0.1 · original submission · Major Revisions

Please carefullly consider the issues raised by the three reviewers. While reviewer-1's main point is a lack of novelty, we are willing to consider confirmatory papers as having intrinsic value, you must address other related studies and place this in better context.

Reviewer 2 has mainly minor points.

Reviewer-3 makes cogent suggestions and I urge you to follow this advice carefully if you wish to resubmit. Please address all comments in your cover letter and a revised paper.

This will need to go back to the reviewers once you have revised it.

Reviewer 1 ·

Basic reporting

No comment

Experimental design

An article on the immunochemiluminescent analysis of the SARS-CoV-2 N-antigen in 2023 looks extremely anachronistic, especially if it is an article from China - in which the pandemic began 1-3 months earlier than in the rest of the world. It is currently well known that serodiagnosis of SARS-CoV-2 infection is most sensitive by detecting antibodies to the N-protein, while the serum virusneutralizing potential can only be assessed by analyzing antibodies to the S-protein, and even better – to its RBD. Detection of the pathogen is the most sensitive and specific by rtPCR from a nasopharyngeal swab, while in the vast majority of cases, the pathogen RNA is determined in the first week of the disease. In fact, the chemiluminescent N-protein detection adds nothing to RNA detection by PCR, other than being much less sensitive.
Exactly confirming my words, the authors showed that the N-protein is determined in the acute phase, much better than in convalescents. The novelty and practical meaning of this study are completely absent.

Validity of the findings

No comment

Reviewer 2 ·

Basic reporting

I am suggesting to add one more reference (see below)

Experimental design

Methods needs some additions (see below)

Validity of the findings

Validity of findings is without any questions, but it needs some clarification (see below)

Additional comments

This is an interesting study, but before publication it needs some revisions:
Major point:

Nucleocapsid antigen is not mutated in COVID-19. Thus the test suggested by authors will also detect other classical type Coronavirus infections. Not to mislead readers this point should be discussed by authors
Minor points:
In Abstract
1. CI – should be spelled
2. ROC-should be spelled
3. Lines 49-51 “In addition, compared with serum antibodies (IgM, IgG, IgA and neutralization antibody (NAb)) (all P < 0.001),there was a significantly higher positive rate of serum SARS-CoV-2 N-Ag in acute COVID-19 patients”

As I could understand the authors are writing about serum antibodies against COVID-19. It should be clarified!

4. Line 122 “ The SARS-CoV-2 antigen and antibody assay reagent kits” catalog numbers should be provided.
5. Line 124 – what are “serum NP antigens”?
6. Study of antibodies against various fragments Spike protein and nucleocapsid antigen in the plasma of convalescent patients revealed gender differences (see “Antibody profiling reveals gender differences in response to SARS-COVID-2 infection “ https://www.aimspress.com/article/doi/10.3934/Allergy.2022002). This paper should be mentioned in the submitted paper.

Reviewer 3 ·

Basic reporting

1. The English language should be reviewed.
Informal expressions were used, for example “What`s more”, in line 55.
And some terms were not appropriate to the sentence, as:
- “Clinical value” in line 34. It is better to use “clinical importance” or “clinical relevance”. As described before “diagnostic value” in line 195.
- “Prevent widespread” in line 63. It is better to use “control”, “monitoring”, because it was a pandemic and diagnostic did not prevent the widespread.

2. Introduction needs to be evaluated. Many references were forgotten. First paragraph (lines 61 to 66) did not refer any literature. There are many papers with information about COVID-19 diagnosis.
3. Pan et al, 2020 (reference 5) have shown that virus inactivation could affect the RT-PCR, but sentence affirm. The term “could” should be more appropriate.
Besides that, Delpuech et al, 2022 (Heat inactivation of clinical COVID-19 samples on an industrial scale for low risk and efficient high-throughput qRT-PCR diagnostic testing) demonstrated it was possible even after virus inactivation. It is necessary to check the sentence “ which “is” affected by sample storage time and virus inactivation temperature”.

4. It is necessary include a paragraph describing the diagnostic methodology chosen in introduction.

5. Poor discussion with only 5 references.

6. The article fulfill complete article structure. Figures and graphs are clear and simple.

Experimental design

SARS-CoV-2 infection course in children and adults is different. I suggest to removing children and teenagers (<14 years old) once there are only 14 patients in this category. And this information was not described in paper. I think data analysis between acute and convalescent without children/teenagers could improve the discussion.

Validity of the findings

The study is interesting, but it did not show an innovation or originality.
As described in line 197, the study reconfirmed data that have already been published.

Even so, depending on discussion, the study could reveal higher relevance. But it is necessary to be robust, strong and polished. It is necessary to discuss about different methodologies to detection and sensibility and its implications. To discuss about detection of nucleocapsid antigen and RNA during the infection course, and about samples collection (swab x blood).

I suggest emphasizing in discussion to improve the paper and to demonstrate its relevance.

Emphasize in discussion about SARS-CoV-2 infection course in < 14 years patients (or remove it) and adults.
It is necessary to better understand methodologies to discuss about CUT-OFF and ROC curve.

Additional comments

Thank you for providing your data, it was necessary to understand some aspect’s study.
I suggest re-analyze data and rewrite the article focusing on a short communication and emphasizing in a robust, strong and polished discussion.

---

## Round 0.2 · Major Revisions

Thank you for revising your manuscript. However, my senior editor colleagues and I feel you need to do a little more to address the concerns identified by reviewer-1 and also by reviewer-3 regarding novelty. While we recognise the value of reconfirming data that is already published, we think that this paper requires more clarity in the discussion to emphasise its relevance. I note the original comments of reviewer-3 here:

'Even so, depending on discussion, the study could reveal higher relevance. But it is necessary to be robust, strong and polished. It is necessary to discuss about different methodologies to detection and sensibility and its implications. To discuss about detection of nucleocapsid antigen and RNA during the infection course, and about samples collection (swab x blood).'

You need therefore to explain and push back reviewer-1's comments regarding the worth of this work at this point in time, and present your view as to why this manuscript is not redundant. Please explain (in a rebuttal AND in the text) what exactly this work brings to the literature.

Reviewer 1 ·

Basic reporting

No comments.

Experimental design

No comments.

Validity of the findings

No comments.

Additional comments

I still remain on the position that in 2023 the work devoted to the SARS-CoV-2 N-antigen serum level detection in COVID-19 patients has niether scientific novelty nor practical value.

Reviewer 2 ·

Basic reporting

no comment

Experimental design

no comment

Validity of the findings

no comment

---

## Round 0.3 · accepted · Accept

Thank you for the further elaboration and justification of your study.